# Usability and Engagement Evaluation of an Unguided Online Program for Promoting a Healthy Lifestyle and Reducing the Risk for Eating Disorders and Obesity in the School Setting

**DOI:** 10.3390/nu11040713

**Published:** 2019-03-27

**Authors:** Martina Nitsch, Tanja Adamcik, Stefanie Kuso, Michael Zeiler, Karin Waldherr

**Affiliations:** 1FernFH Distance Learning University of Applied Sciences, Ferdinand Porsche Ring 3, 2700 Wiener Neustadt, Austria; martina.nitsch@fernfh.ac.at (M.N.); tanja.adamcik@fernfh.ac.at (T.A.); stefanie.kuso@fernfh.ac.at (S.K.); 2Department for Child and Adolescent Psychiatry, Medical University of Vienna, Waehringer Guertel 18-20, 1090 Vienna, Austria; michael.zeiler@meduniwien.ac.at

**Keywords:** usability study, online health intervention, adolescents, school setting, eating disorders, overweight, prevention, engagement, E-Mental Health

## Abstract

Implementing integrated online prevention to reduce the risk of both obesity and eating disorders, in the school setting, is a promising approach. The challenge is to develop highly user-friendly and motivating programs, to foster adherence and effectiveness. The purpose of this study was to evaluate the usability of such a universal prevention program for students aged 14–19 years, and to address engagement issues. A mixed-methods approach was chosen, consisting of a think-aloud task, a semi-structured interview, and a questionnaire including items on sociodemographic characteristics and the System Usability Scale (SUS). Usability tests were conducted in two rounds, with five adolescents participating per round. Mean score in the SUS was 92.5 of 100 points (range 85–100), in the second round, after some adaptations from the participants’ feedback. In the course of the think-aloud tasks and interviews, five major themes emerged—visual design, navigation, mode of transfer, content, and engagement conditions. Interesting headlines, gamification, and monitoring tools are crucial for engagement. Apart from the importance of using the program during school hours, the study showed that problems currently perceived as important by the target group, need to be considered and addressed, prior to offering them prevention programs, which highlights the importance of a user-centered design.

## 1. Introduction

Prevalence rates of overweight and obesity, worldwide, have increased drastically [1,2]; consequently, the World Health Organization has labeled obesity as a “global epidemic” [3]. Over-evaluation of weight and shape, restricted eating and dieting are also on the rise [4], and are amongst the strongest risk factors for full and sub-threshold eating disorders (EDs) [5]. Whereas, EDs like anorexia nervosa and bulimia nervosa especially affect girls and young women, health implications of problematic eating and exercise habits affect both women and men, at all ages [4,6,7]. Recurring episodes of unhealthy dieting, binge eating, and purging behavior are also common in adolescents suffering from overweight and obesity [7,8,9], influenced by the social pressure on overweight people to reduce their weight. However, a couple of longitudinal studies have shown an association between dieting and weight gain, among adolescents [10,11,12,13].

Given the shared risk factors for eating disorders and being overweight, researchers have called for integrated approaches to prevention [14]. Such approaches are also considered to reduce the risk of unintentionally causing eating disorders with obesity prevention programs [14], and they have the potential to reduce stigma. There already exists a number of effective prevention programs addressing the full range between anorexia nervosa and obesity [15].

Due to their scalability, their potential to reach a large number of people, and relatively low costs, online programs seem to have a high potential to tackle this public health challenge [16]. Especially in group settings, like school classes, online programs offer the possibility to provide different program parts for different participants, simultaneously, within the same school class, thus allowing to tailor program content to participants’ characteristics, like gender, and risk status [17]. Furthermore, it is assumed that adolescents, especially, can benefit from Internet-based interventions [18].

More than 95% of European adolescents and young adults between 16 and 24 years, are using the Internet regularly, therefore, Internet-based or mobile phone applications/platforms could be a key resource in providing health information to adolescents [19]. As for interventions promoting mental health, there is evidence of positive effects on young people’s mental health, especially when implemented in the school setting [20,21,22]. However, the challenges of such intervention programs are poor adherence and high dropout rates [23,24]. Since these factors limit the effectiveness of Internet-based interventions, further insights into user technology interaction are urgently needed. Traditional approaches like randomized controlled trials (RCT) are not appropriate for investigating those complex phenomena, including many contextual and confounding factors. Thus, several authors suggest the application of a mixed-methods design, focusing on process variables like dropout and usage, and outcome variables like costs, health condition, or adherence [25].

Accordingly, this study aimed to evaluate the usability of a school-based online intervention program for adolescents, as well as to address engagement issues, to reduce dropout rates in the “Healthy Teens @ School” study, a multi-country cluster RCT [26]. The intervention was adapted from an evidence-based program developed in the USA, called “Staying Fit” [17,27]. The unguided online intervention program “Healthy Teens @ School” aims to promote a healthy lifestyle and reduce problematic eating behavior, eating disorder risk, and obesity risk, among adolescents aged 14 to 19 years. Prior to the conduction of the main study on “Healthy Teens @ School” in Austria, we conducted three focus groups with representatives of the target group [28], and a subsequent usability study. For the usability study we used a mixed-methods approach suggested by Nitsch et al., [29] which proved to be an appropriate design, in order to get detailed insights into users’ needs and to learn more about the challenges regarding engagement at the same time. In this context, especially the evaluation of the first program modules and the users’ first interaction with the program, are crucial elements for determining future engagement and adherence patterns, since most participants drop out from online interventions at this very early stage of a program [23,30,31]. This paper will present the results of the usability study and highlight the importance of investigating engagement issues, prior to the program start.

## 2. Materials and Methods

### 2.1. Recruitment

Potential participants were recruited via social media (Facebook). Girls and boys aged between 14 and 18 years, from different schools in Vienna, were invited to take part in a study aiming to test the prototype of an online program for promoting healthy habits in pupils. The inclusion criteria for this usability study reflected the target population of the “Healthy Teens @ School” program that was designed for students between 14 and 19 years. Participants were offered gift cards of € 20 for their participation.

Following the principles of usability evaluation, we conducted usability tests in two rounds, with five participants per round. A sample size of five participants per round was regarded as sufficient to detect more than 85% of usability problems. Including more participants would have required more resources in terms of time and money, while only producing repetitive information [32,33]. In the first round, five participants tested the prototype of the program on the computer. Based on the results of the first round, the program was modified and tested by another five participants in the second round.

### 2.2. Program/Intervention

“Healthy Teens @ School” is a ten-week online program (one online-module per week) designed to promote a healthy lifestyle and reduce eating disorder risk and obesity risk. Based on a screening questionnaire, which includes the assessment of eating behaviors, eating disorder risk, weight status, weight/shape concerns, physical activity habits, stress coping, depression, anxiety, self-esteem, and quality of life, the participants get access to the online program and are assigned to either one of two program tracks. Adolescents with normal weight (with and without eating disorder risk) are assigned to the “Healthy habits” track, overweight adolescents (>85th sex-age-specific BMI percentile) are assigned to the “Weight management” track. In ten modules, the students learn about building a healthy lifestyle, about balanced nutrition and physical activity habits, about ways of improving their body image and body satisfaction, as well as ways of improving media literacy. Whereas the content in the “Healthy habits” track is framed towards building a healthy lifestyle, the content in the “Weight management track” is framed more towards maintaining a healthy weight. In the course of the three preceding focus groups, especially stress in school turned out to be a major issue for the students [28]. As a result, we adapted the program and added elements about how to deal with difficult and challenging situations, emotions, and stress, in order to prevent mental health problems. Details on the program content are published elsewhere [26]. The program is based on principles of Cognitive Behavioral Therapy (e.g., goal setting, behavior monitoring, practical exercises to be tried out between online modules). The users are encouraged to use a “diary”-function that can be accessed via PC or an app, to monitor their habits on a daily, or at least weekly basis. Furthermore, the users receive feedback on their strengths and weaknesses, based on the results of the screening questionnaire.

The program is provided on an online platform hosted by Minddistrict GmbH. The users log on the platform via an e-mail address and a self-chosen password. From the user’s dashboard, the screening questionnaire and program modules can be accessed; students can see new tasks assigned and they have the possibility to contact the study team via a messaging function. Screenshots of the user’s dashboard and the program module including the app-diary are provided in Figure 1 and Figure 2.

The program can be accessed within and outside school lessons, via different devices, like computers, tablets, and smartphones [26]. For this usability study, we used a shortened version of the screening questionnaire and feedback, the first two online modules, as well as the diary function. We decided to test those parts that are scheduled at the beginning of the program, as they may be crucial for subsequent adherence [23,29,30,31]. The screening questionnaire used in the usability study included about 60 items, covering sociodemographic questions (including height and weight) and standardized questionnaires, to assess stress coping, intuitive eating, self-esteem, and health-related quality of life. The KIDCOPE [34] assesses the frequency and perceived effectiveness of ten different coping strategies for a defined problem. Coping strategies are divided into active strategies (e.g., cognitive restructuring), avoiding strategies (e.g., social withdrawal) and negative strategies (e.g., self-blame). Prior to rating the frequency and effectiveness of the used coping strategies on a 4-point scale, participants were asked to define a situation or problem that has stressed them often. Coping is conceptualized as a dynamic process and not a stable personality trait, resulting in a rather limited re-test reliability, after 10 weeks (r = 0.15–0.43). The Intuitive Eating Scale (IES, [35]) measures the individuals’ tendency to follow their physical hunger and satiety cues, by determining when, what, and how much to eat. The 23 items are rated on a five-point scale and are summed up to a total score and four subscales (“Unconditional Permission to Eat”, “Eating for Physical Rather than Emotional Reasons”, “Reliance on Hunger and Satiety Cues”, and “Body-Food Choice Congruence”). Good internal consistencies of the total score (Cronbach Alpha 0.91 for females and 0.82 for males) and subscales (Cronbach Alphas >0.72) were reported [36]. The Rosenberg Self-Esteem Scale [37] is a standard measure for self-esteem, consisting of ten items rated on a five-point scale. Internal consistencies are high (Cronbach Alpha = 0.88) [38]. Finally, we also used the Inventory of Life Quality for Children and Adolescents [39], which measures subjective well-being and satisfaction in seven different domains, including school, family, peers, leisure activities, physical health, psychological health, and overall health-related quality of life. The seven items are rated on a five-point scale and calculated to an overall score. The re-test reliabilities for the total score reported by the authors range from r = 0.60 to r = 0.80.

### 2.3. Procedure

We conducted an evaluation study focusing on aspects of usability and engagement, following the study protocol of Nitsch et al. [29].

Prior to testing, informed consents of the participants and legal representatives were obtained, and the researcher explained the procedure. The usability testing sessions consisted of three parts, (a) program use while performing the think-aloud technique, (b) a semi-structured interview, and (c) completion of a questionnaire, including sociodemographic questions and the System Usability Scale (SUS, [40]). The think-aloud process involved four tasks. First, registration on the platform and setting a password; second, completion of the online assessment and reading the feedback; third, completion of the first and the second module of the “Healthy Teens @ School” program; and fourth, to make a diary entry on the PC and via the app. Before starting the think-aloud task, the students practiced the technique by completing an Internet-based task (searching for a DVD on Amazon), which was unrelated to the “Healthy Teens @ School” program. The moderator of the test presented the different think-aloud tasks to the participant. However, participants were instructed to act as if they were alone, talking to themselves. Participants were encouraged to express anything that came into their mind, including positive and negative thoughts about the program, as the researchers on-site were not involved in the development of the original program but are only interested in improving it. There was little interaction with the researchers, only in case the participant stopped thinking aloud, the moderator motivated the student to try again. All comments and problems during the think-aloud task were recorded and protocolled by an observer. In addition, the computer screen was recorded (©Microsoft Expression Encoder 4 Screen Capture SP2), in order to capture verbal and non-verbal reactions of the participants.

Following the think-aloud task, a semi-structured interview was conducted. The interview guide included questions about expectations towards the program, impressions of the program, questions about program content, the diary, motivation for completing the program, and if the participant would recommend the program to others.

Subsequent to the semi-structured interview, the participants were asked to complete questions on sociodemographic characteristics, their current Internet use, as well as the SUS questionnaire [40]. The SUS is a method for measuring the usability of an application. It is a standardized 10-item questionnaire. The items are rated on a five-point Likert scale. A high reliability (Cronbach Alpha = 0.91) was reported for the total SUS score [41].

The usability testing sessions took 55 to 90 min, in which participants had the possibility to familiarize themselves with the program and answer the questions. The testing sessions were conducted from January till February 2017, and took place at the FernFH Distance Learning University of Applied Sciences in Vienna and at the Medical University of Vienna.

The study protocol was approved by the Ethics Commission of the Medical University of Vienna (Austria, Record Number: 2209/2015).

### 2.4. Analysis

Both, the think-aloud task and the interviews, were videotaped and audiotaped, then transcribed verbatim, including the non-verbal reactions of the participants, based on the video recordings and the observer notes. The transcripts were coded and analyzed, using NVivo 11. Thematic analysis [42] was used to identify new aspects and relations between the themes and to combine data from the think-aloud task and the semi-structured interviews. The themes and categories were discussed, reviewed, and interpreted by the research team. The SUS questionnaire was analyzed in © Microsoft Excel, as follows. After recoding the inverted items, item codings were summed up and the sum score was multiplied by 2.5. This resulted in a total SUS score that ranged from 0 (no usability) to 100 (perfect usability). Finally, we calculated an average score for each usability round. A SUS-score of higher than 70 points is regarded as an acceptable usability (representing the average SUS scores found in other usability studies) and scores higher than 85 points are regarded as excellent usability [41].

## 3. Results

### 3.1. Participants

All ten participants were students between 14 and 18 years old (mean age = 16.2 years, standard deviation (SD) = 1.2), attending different high schools in Vienna. In the first usability test round, four male students and one female student took part, and in the second round, four female students and one male student participated. Their BMI ranged from 18.8 to 24.2 (mean = 21.67, SD = 1.73). None of the participants had a diagnosed psychiatric disorder or received psychotherapeutic treatment. In the IES which was defined as the main outcome of the intervention, eight participants had average scores and two participants had above-average scores (mean score = 3.76, SD = 0.45).

### 3.2. SUS-Questionnaire

Regarding the SUS-questionnaire, the usability of the program was in the acceptable range (SUS-Score > 70) in the first round, and in the excellent range (SUS-Score > 85) in the second round. The average SUS score was 84 (range from 67.5 to 97.5), in the first round, and improved to an average SUS score of 92.5 (range from 85 to 100) of 100 points, in the second usability test round.

### 3.3. Think-Aloud Task and Semi-Structured Interview

By using the think-aloud tasks and the semi-structured interviews, we were able to identify five central themes, referring to visual design, navigation, mode of transfer, content, and engagement conditions.

#### 3.3.1. Visual Design

Most of the students liked the visual design of the “Healthy Teens @ School” program, especially the pictures and graphics were highlighted as positive. The layout of the program was perceived as *“youthful”, “cool”, “pretty”, “appealing”,* and *“clearly arranged”* and reminded some participants of social media platforms and search engines: *“The landing page looks up-to-date, similar to social media and Facebook and instant messaging design, appealing to teenagers (...) I also liked the layout and pictures”* (male#5). The program’s color design was rated ambivalent by the participants. Some liked the variation, others recommended a reduction in colors or a more targeted use of colors. *“Maybe, there should be less red, because I know, especially on a bright website, red indicates that there is something very important or wrong”* (male#7). Some students observed, that the font size was too small in some areas. All participants mentioned, that they like the layout of the diary, regardless of using it on the computer or via the app. They were also motivated to complete the diary because they liked the visualization of their improvement in healthy habits over time. Furthermore, participants liked the absence of advertising elements in the program.

#### 3.3.2. Navigation

The majority of the students described the navigation of the program as “*easy*” and “*self-explanatory*”. However, it seemed to be a challenge to create a secure password. Only one of ten participants managed to include all demanded aspects (minimum 12 characters and one special character) at the first attempt.

In the think-aloud sessions we observed some challenges finding the next module or the diary function. Some asked for help, others found the next module or the diary after repeatedly trying. Consequently, we produced and included a tutorial video lasting five minutes for the second usability test round. This video included a short introduction to the navigation and the functions of the program and the platform. Most participants liked the tutorial video, they described it as “*helpful*”, “*reasonable for orientation*”, and “*well-illustrated*”. Others said, that they would not need such a video, stating young people prefer to try out programs and do not want to watch videos about the navigation: *“I think, it is self-explanatory (…). And I think, most teenagers would prefer to simply try it out instead of watching some explanation for a long time. Okay, actually, it’s not so long, but I think, that most people would just try it out”* (female#6). After showing the tutorial video in the second usability round we could monitor more targeted approaches to find the diary, the different modules and an overall improvement in the platform navigation. A participant even mentioned that “*you will instantly find your way”* (male#7) in the program.

#### 3.3.3. Mode of Transfer

This theme refers to the way the content and information is communicated to the user. After completing the screening questionnaire, some participants were irritated by certain questions of the IES, which sounded similar and felt like a repetition to them.

During the first test round, participants had to scroll to the top several times, as due to its length, the response options of the IES questionnaire did not fit on one screen, after a certain number of items. Therefore, prior to the second test round, we divided a long block of questions into two smaller parts, in order to ensure that the response options were visible at all times. Negated questions were harder to understand for some students. One person said that sometimes the wording in the IES questionnaire was unclear (e.g., “substantial food”) or students indicated questions that appeared strange to them. Furthermore, some participants mentioned that the open question asking to describe a personal stress inducing problem (out of the KIDCOPE-questionnaire), was somewhat confusing, since the subsequent set of questions, directly referring to the mentioned problem, were not always fitting.

In the interviews, participants criticized the long text passages, very detailed information and suggested shorter texts and sentences: *“There was a bit too much written text, but the texts were not always the same, and addressed you directly”* (male#3). An aversion for long texts became also apparent in the think-aloud sessions, where many participants only skimmed or even skipped longer texts. On the contrary, interactive tools like quizzes and questions were always noticed and used and rated as positive: *“It is better than just reading tips, (…) that you can fill out things and instantly get feedback*” (male#1). Accordingly, the students underlined the importance of the “gamification” of the program; they liked to do quizzes to repeat the content or knowledge games, instead of reading long texts with detailed information and complex wording. For example, one girl argued that interactive tools might also be helpful with regard to concentration: “*You can’t finish it [the program] in five minutes; instead you really have to take your time, sit down and read it in detail. Maybe it should be a bit shorter, so that you don’t have to read that much, yeah, more quizzes, and just make it simpler, so you don’t have to concentrate so hard*” (female#2). Furthermore, the participants said that they liked to explore new things in the program and that they like being addressed personally with “you” or “your”. Since text passages without interesting headlines were partially only skimmed and students stated that significant headlines motivated them to read text passages, we added more headings after the first round.

The diary tool was considered as interesting and useful, especially regarding the possibility to visualize habits over time. In the second round, the diary function was available on the computer and additionally on the app. Both modes were rated as positive, but two advantages of the app were highlighted—the app appeared more game-like and chances of using the app more frequently were rated higher, since it is more convenient to use the program on the smartphone: *“It is done real quick, it takes less time than a normal diary entry. Well, on the smartphone it is quick, you can do it on the subway or on the way home, that’s fine, that’s convenient”* (male#7).

#### 3.3.4. Content

In general, the students denoted the information delivered in the different parts of the “Healthy Teens @ School” program as “*helpful*”, “*informative*”, “*diversified*”, “*good to read*” and “*meaningful*”. One participant said that the program “*is cool”* because “*it is not counseling, not like at the psychologist or the doctor, you don’t feel sick (...) it is a kind of help but not in an uncomfortable way (…) not bad for your self-worth*” (female#8).

The explanation and the outcome of the feedback were described as fitting to the person and very informative: “*I think, it fits me well, and it came across very personal and that I find really cool, I mean that was really cool. And for me personally, it is real fun to fill out quizzes and to get such feedback. That is fun, that is cool*” (female#6).

The provided information in the modules was described as meaningful and interesting. Some students expressed that they have already heard about some of the information, but it was also nice to hear familiar content again. Some parts of the content were new to them. For example, most participants stated that they already knew about the food pyramid or learned about it in school. The information on how many portion sizes a day is recommended for the different food groups, and how portion sizes can be measured, was perceived as new and interesting: “*I didn’t know much yet, for example the daily food requirement, how many snacks you should eat, what and how much food you should eat to be balanced, in school and at home*”(male#5).

#### 3.3.5. Engagement Conditions

The participants stated that they would use the program in their daily routine, if they feel that they need it and would recommend the program to colleagues and friends, if they had psychological or health problems or if they think the content would be interesting for them. Some students indicated that they like the program because it was informative, exciting, easy and fast to use. The motivation to complete the program was the desire for a change in daily lifestyle and the personal feedback received. For example, one participant stated that “*if you do that (the program), then you want that something changes. If you wouldn’t want to change something, then you would not do it*” (male#3). The possibility to use the program, including the diary, on different devices like computer, smartphone, and tablet was very attractive for the users and might also improve the adherence of participants. Especially, the smartphone version of the program was described as “*user friendly*”, “*comfortable*” and “*always within reach*”. For example, one participant said that she would use the program “*more likely on the smartphone as it’s always in reach. And I do not have to turn on the computer, enter the password and so on (…) but your smartphone is always there. You can also do it on the way*” (female#9). Another participant found that he personally would prefer doing the program “*on the computer because there is a keyboard and I prefer that, but for many others at my age the smartphone is the number one and they would prefer doing it on their smartphone*” (male#5).

Since a lack of time also seemed to be an important aspect for students, they positively highlighted the length of the program modules. The majority of the participants indicated that over the course of 10 weeks, investing 20 to 30 min once a week, seems to be feasible.

The possibility of using the program within school lessons turned out to be an important factor in terms of engagement. On the one hand, all participants stated, that if they had the chance to do the program in class, they would use it regularly. However, in most cases the program simply turned out to be a better alternative than regular school lessons: *“I would do it, I’m not sure if the other students would do it as well, but when it takes place in school instead of a regular class, then I think they would. It depends on how the first five modules are, if I would do it at home as well*” (female#10). On the other hand, although the school was generally described as an adequate setting for using the program, the school setting itself raised many issues in connection with the topic of stress. For example, in the course of the usability session, students were asked to describe a recent problem that bothered them and nine out of ten participants gave an example of school-related stress, including “*work overload”, “exam stress*”, stress related to bad grades and interpersonal problems with classmates. During the think-aloud task, many participants emphasized consequences of school stress, for example influencing their overall well-being: *“I’m not feeling well at the moment, because I have school stress”* (female#10). Another participant highlighted that stress has a negative effect on his leisure activities: “*I have too much schoolwork and don’t have much free time to do anything else*” (male#3). This finding is especially important, since it validates and highlights the results of the pre-study, in which focus groups with students were conducted and stress turned out to be a major issue as well [28]. One participant emphasized that she “*would recommend [the program] to students experiencing stress or health problems; but rather not to others*” (female#10).

## 4. Discussion

The aim of this usability study was to evaluate and advance the usability and engagement aspects of an unguided online intervention program for promoting a healthy lifestyle and to reduce the risk for eating disorders and obesity among adolescents. Additionally, we aimed to validate the results of preceding focus groups with representatives of the target group. The application of a mixed methods research design not only allowed us to gain deeper insights regarding the students’ perception of online health promotion and prevention programs and their media usage, but also to improve and test the program in a very effective way. In this usability study, we found five major themes—visual design, navigation, mode of transfer, content, and engagement conditions.

Especially, the findings of the current study regarding the design, navigation, and the mode of transfer of online interventions are comparable to other studies. For example, one major finding was the fact that the participants disliked reading long texts and preferred multimedia content, which was also a result in similar studies [29,43,44]. This implies that the transfer of content on prevention, via multimedia elements, might increase engagement and, therefore, is preferable to text-based information [43,45]. Furthermore, instead of presenting participants an introductory text about navigation and functions of the program, we produced a short tutorial video, in order to reduce the amount of text. Although some participants doubted the need of the tutorial video for program navigation, we observed an overall improvement regarding the navigation of the different tools, after its implementation.

Since some participants tended to skim or skip long texts, it is crucial to visually distinguish important from less important text parts [44]. We tried to highlight important information, concerning content and navigation, by putting it into boxes with prominent colors, like red and yellow. Informative headings facilitate orientation and let users choose what to read. In general, this might be a more realistic approach regarding the needs and usage habits of adolescents, than to expect them to read everything.

Similar to the study of Nitsch et al. [29], some participants expressed objections about the length of the assessments and particular questions. As a result of the standardized instruments used, a change of the questions was not possible. Based on present and previous research, there is clearly a lack of user-friendly research assessment, which have been developed to use within online programs, by considering adequate wording and format. However, a very positive mentioned tool, in conjunction with the assessment, was the subsequent personalized feedback, which all participants regarded as suitable for their needs and as very informative. Especially, the prospect of receiving a feedback turned out to be an important motivational factor to complete the assessment.

Furthermore, especially for adolescents, the integration of interactive and gamification elements seems to be an essential prerequisite [46]. Changes in the layout, navigation, and content showed improvements in the second usability round.

We observed that time was a crucial factor for using the program or recommending it to others. The length of the weekly modules was described as suitable and the daily diary was evaluated as very good and realistic, regarding regular program use. Accordingly, the participants mentioned the importance of different user devices. The possibility to use the online intervention program, including the diary, on the computer, smartphone, and the tablet increases the accessibility.

Another motivating point was that the program addressed the users personally with “you”. In the diary the participants liked the possibility to visualize habits over time and mentioned this as a factor for using the diary regularly or daily.

A crucial factor in terms of engagement and regular program use was the possibility of working with the program in the school setting, and not in the free time. This result can also be confirmed by various other studies, for example Neil et al. [47] highlighted that participation in an online prevention program in the school setting, yields higher adherence than not participating in the school. Additionally, the majority of youth can be reached through the school setting and since schools are familiar settings for students, stigma, for example, associated with mental health interventions can be reduced [48]. Generally, evidence also suggests that the school is an important setting for universal prevention interventions [49,50].

However, when implementing interventions in particular settings, confounding or contextual factors concerning the setting itself need to be considered as well. The present study showed that the school setting raised various issues related to the topic of stress. For example, when students were asked about a recent problem, the majority of the participants mentioned a school-related problem. Accordingly, the problems currently perceived as important by the target group, need to be considered and addressed, prior to offering them programs for health-related issues, which they might not even think of, or are interested in. The results of this study, as well as our previous research on engagement issues, suggest that framing a program, primarily designed to reduce eating disorder and obesity risk, towards stress and stress reduction, might help to improve engagement. Stress turned out to be a major issue for adolescents who participated in our study, and this might also motivate them to use the program. Furthermore, program content covering a balanced nutrition, physical activity, and body image, can easily be linked to the topic of stress, for example, by highlighting the association between high levels of stress and fast eating or the benefits of regular physical activity for stress reduction. On the other hand, providing adolescents with helpful coping strategies, might prevent them from resorting to negative behavioral patterns, like self-blame or emotional eating. Furthermore, participating in a program labeled as a stress-reduction program might also be less stigmatizing than participating in a program for preventing eating disorders and obesity, since almost every adolescent experiences stress in some way or another. This was also an issue in another study where adolescents said that the terms describing mental disorders sounded daunting. They recommended that the program title and content should be framed positively (e.g., “improving healthy habits” instead of “reducing risk for eating disorders and obesity”) [28]. Generally, we suggest evaluating the potential users’ needs, prior to the program start, since addressing their needs might contribute to adherence and engagement. However, for students in a different context, other topics might be more relevant.

In this context, the value of user-centered design needs to be highlighted, since it does not only allow us to tailor interventions to the users’ needs, but also offers a high potential to improve the translation of evidence-based health research into particular settings [51]. Moreover, the design of online (mental) health programs affects their uptake and use [25,52,53,54]. Accordingly, future research needs to address this gap, by highlighting the value of integrating aspects from different interdisciplinary fields. In this context, especially a combination of different methods from the broader field of implementation science and human computer-interaction, is suggested [52,53].

Besides the usability and engagement issues, which were addressed in the present study, further steps are necessary to ensure a successful and sustainable implementation of online prevention programs in school settings. Strong evidence for the effectiveness and cost-effectiveness of such programs are an important prerequisite for long-term implementation, not only according to various stakeholders in the school setting, but also from an ethical point of view [28]. Therefore, the “Healthy Teens @ School” program is currently being tested for its effectiveness and cost-effectiveness, as part of a cluster RCT, involving more than 800 adolescents in two European countries [26,55]. Since adherence to an online prevention program and consequently its effectiveness are not only affected by usability issues but also by other contextual factors, such as participant, setting, and intervention characteristics, we also aim to investigate a variety of moderators and mediators potentially influencing outcome [56]. Furthermore, we aim to evaluate the potential public health impact of this program by applying the Reach-Effectiveness-Adoption-Implementation-Maintenance (RE-AIM) framework [57]. RE-AIM encourages the researcher not only to focus on effectiveness but also to obtain a number of indicators—reach of adolescents, adoption (willingness of schools to offer the intervention to their students), implementation (e.g., adherence, adaptations to be made), and maintenance (both on the individual and organizational level). Overall, these research steps will help us to develop an online prevention program for EDs and obesity, which is well-accepted by schools and adolescents and which can be further disseminated to a wide range of schools.

### Limitations

The participants completed only the first part of the assessments and the program modules. Therefore, results can only partly be generalized to the other assessment questionnaires or modules. However, the evaluation of engagement and usability issues of the users’ first interaction with the program proved to be a suitable method, in order to determine future participant behavior [29]. Moreover, most participants drop-out during the very early stage of an online program, which indicates that the first impression is crucial for future adherence [23,30,31]. Furthermore, since the usability testing took place in a lab, it is possible, that the students were influenced by the situation itself. However, we addressed this issue and gave the participants the opportunity to test the think-aloud method, before we started with the actual study.

Not all of the feedback of the participants could be implemented in the second round and subsequently for the actual clinical trial, due to multiple reasons. Although much of the criticism about some questions seemed legitimate, the used questionnaires were standardized questionnaires that could not be changed, simplified, or shortened. Some problems or barriers were related to the navigation which was not intuitive in some aspects and led to the need to try out a few times. Since our possibilities to change these aspects, which were linked to the functions of the Minddistrict platform, were limited, we tried to tackle these issues by providing the tutorial video and additional information on the navigation in the text.

Another major limitation might be the gender imbalance of the test rounds. In the first round, four boys and one girl participated and in the second round four girls and one boy took part. We tried to reach a balance of gender per round, but due to time collisions and short-term illness of two participants, this was not feasible.

## 5. Conclusions

This usability and engagement study was conducted in preparation for a randomized controlled trial of the unguided online program “Healthy Teens @ School”, taking place in the school setting. This study helped us to improve the usability of our online program, prior to the program start, in five major areas—visual design, navigation, mode of transfer, content, and engagement conditions. Furthermore, the results of this study contribute towards a wider research base, showing that usability and engagement issues in this first phase of the users’ interaction with the program is not only inextricably linked, but is essential for identifying future engagement and adherence issues. Especially, the users’ current needs have to be evaluated and addressed. As the results of this study showed, the engagement issues of this particular target group were closely linked to specific setting-related issues, such as stress in school or the possibility to use the program during school hours. This implies that the involvement and participation of the target group in the development of online interventions, should be an essential part in future research designs. Especially for prevention interventions, intrinsic motivation plays a major role, since members of the potential target group usually do not feel a certain urge or level of suffering that motivates them to use an online program. Additionally, from a public health perspective, it is important to point out the benefits of such programs for the target group. In this context, user centered-designs do not only help to improve the usability and adherence of online programs but also their overall effectiveness.

## Figures and Tables

**Figure 1 nutrients-11-00713-f001:**
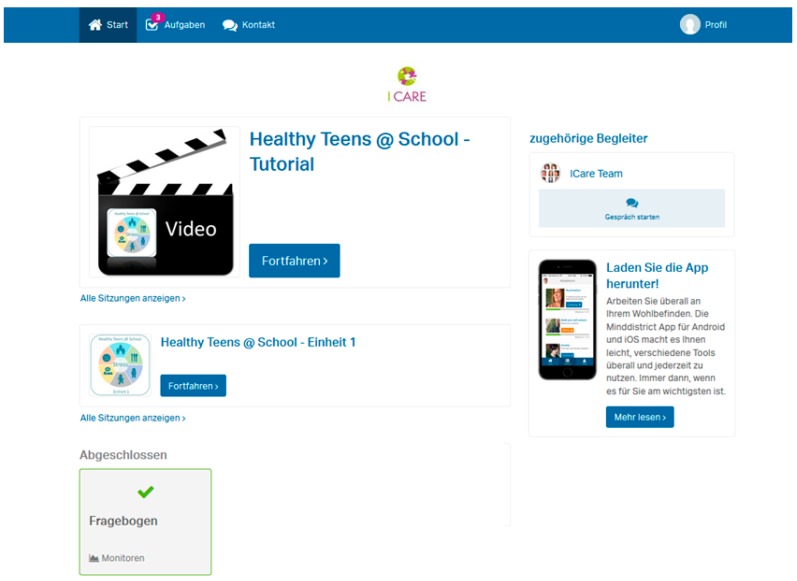
Screenshot of the user’s dashboard on the online platform.

**Figure 2 nutrients-11-00713-f002:**
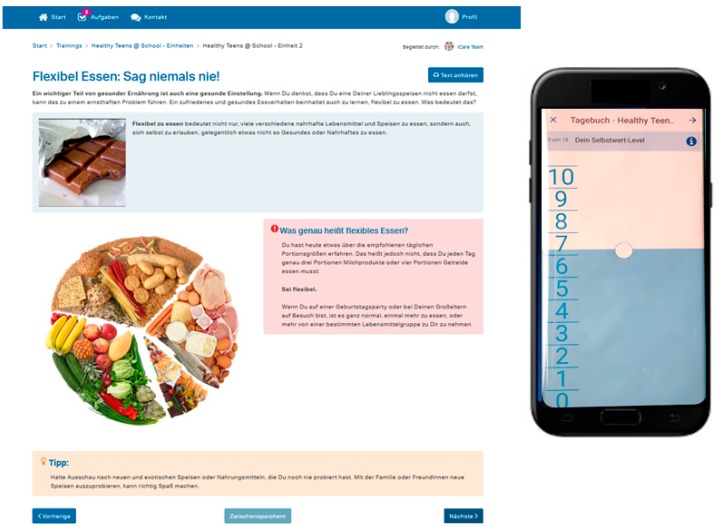
Screenshot of one page in module 2 of the Healthy Teens @ School program and the app-diary.

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
