# Peer review of "Usability and Engagement Evaluation of an Unguided Online Program for Promoting a Healthy Lifestyle and Reducing the Risk for Eating Disorders and Obesity in the School Setting"

_nutrients, 2019, doi:10.3390/nu11040713_

Reviewer 1 Report

This is a well-presented manuscript on an extremely interesting and topical subject. The paper is likely to be well-read in the field of e-health.

MAJOR

1.     As the study used qualitative analysis, it would be important to include some tables or figures or quotes to support the conclusions made by the study. For example, a table showing the themes and sub-themes that emerged in the analyses, as well as quotes that support those themes / sub-themes.

2.     Around line 300, there is a suggestion that use of multimedia in the app may be important to end user engagement. However elsewhere in the manuscript is a statement that videos were not universally welcomed by users. These two pieces of information seem contradictory, and so it would be helpful to reconcile them. For example, is it that multimedia may be welcomed in the app, provided that any videos provided are not longer than x minutes?

3.     Can the Authors be more explicit in articulating their ideas around line 345? This is a very important point. Are the Authors saying that if one wants to use an intervention to prevent eating disorders and manage weight in students / young people, then it would be better to administer the intervention within the context of a stress reduction program, because that is what the students / young people are most concerned about, and most motivated by?

4.     The Think Aloud exercise is an excellent idea. How did the Authors help to ensure that the students / young people felt comfortable enough to say what they really thought of the app? For example, how did they mitigate the possibility that the students / young people may have been trying to say only nice things about the app, and hiding any true negative feelings? More detail about this would be beneficial in the paper.

5.     The manuscript should finish with a short conclusion, and not finish with the limitations.

MINOR

6.     The word ‘appealed’ at line 246 should probably be ‘appeared’?

7.     At line 299 there is a typographical error (should be ‘of’ not ‘auf’).

8.     The word ‘legit’ at line 357 should be replaced with a non-slang word, such as ‘legitimate’.

Author Response

Response to Reviewer Comments

We would like to thank the reviewers for their valuable comments and suggestions. After completion of the suggested edits, we believe that the manuscript has considerably improved. Please find below a description of how each comment was addressed in the manuscript. (Changes compared to the original submission are highlighted in the manuscript.)

Reviewer 1:

This is a well-presented manuscript on an extremely interesting and topical subject. The paper is likely to be well-read in the field of e-health.

Thank you!

MAJOR

 1.     As the study used qualitative analysis, it would be important to include some tables or figures or quotes to support the conclusions made by the study. For example, a table showing the themes and sub-themes that emerged in the analyses, as well as quotes that support those themes / sub-themes.

Response 1: Thank you for this comment. We added quotes in the results section to illustrate the emerging themes. We decided to include the quotes directly in the main text instead of putting them into a separate table because we thought this is more reader-friendly. All quotes are in italic letters, so that they can be easily distinguished from the other text passages.

2.     Around line 300, there is a suggestion that use of multimedia in the app may be important to end user engagement. However elsewhere in the manuscript is a statement that videos were not universally welcomed by users. These two pieces of information seem contradictory, and so it would be helpful to reconcile them. For example, is it that multimedia may be welcomed in the app, provided that any videos provided are not longer than x minutes?

Response 2:  Thank you for your comment on this important point. Multimedia content including videos to transfer the content of a prevention program was generally welcomed by the users. However, the tutorial video was introduced to explain the platform navigation and not to transfer program content. Thus, some users doubted its use. We restructured this paragraph in the discussion section to make this distinction clearer.

3.     Can the Authors be more explicit in articulating their ideas around line 345? This is a very important point. Are the Authors saying that if one wants to use an intervention to prevent eating disorders and manage weight in students / young people, then it would be better to administer the intervention within the context of a stress reduction program, because that is what the students / young people are most concerned about, and most motivated by?

Response 3: Thank you for this question. We have added a few sentences to explain our ideas about this point. Indeed, with respect to the results of this study and our previous research, we think that - with regard to engagement issues – it may help to frame a program designed for reducing eating disorder and obesity risk towards stress and stress reduction. Stress turned out to be a topic which adolescents are most concerned about and this may also motivate them to use the program. The program content referring to balanced nutrition, physical activity and body image can also be easily linked to the topic of stress, for example by highlighting the association between high levels of stress and fast eating or the benefits of regular physical activity for stress reduction. On the other hand, providing adolescents with helpful coping strategies might prevent them from resorting to negative behavioral patterns, like self-blame or emotional eating. Furthermore, participating in a program labeled as stress-reduction program may also be less stigmatizing than participating in a program “for preventing eating disorders and obesity” since almost every adolescent experiences stress. This aspect also came up in another study where adolescents said that terms describing mental disorders look daunting and the program title and content should be rather positively framed (e.g. “ improving healthy habits” instead of “reducing risk for eating disorders and obesity”) (Zeiler et al., 2019 under review)

4.     The Think Aloud exercise is an excellent idea. How did the Authors help to ensure that the students / young people felt comfortable enough to say what they really thought of the app? For example, how did they mitigate the possibility that the students / young people may have been trying to say only nice things about the app, and hiding any true negative feelings? More detail about this would be beneficial in the paper.

Response 4: Thank you. We agree that this is an important point. During the instruction of the think aloud task, we explicitly encouraged the participants to express anything that comes into their mind including positive and negative thoughts about the program. Additionally, we mentioned that the present researchers were not involved in the development of the original program but are interested in improving it. Thus, negative thoughts and opinions would not hurt any feelings of the researchers but will help them to improve the program. We added more details on the instruction of the think aloud task in the “procedure” section of the manuscript.

5.     The manuscript should finish with a short conclusion, and not finish with the limitations.

Response 5: Thank you for your comment on this important point. We have added a conclusion at the end of the manuscript (see also reviewer 2).

MINOR

 6.     The word ‘appealed’ at line 246 should probably be ‘appeared’?

 Response 6: Thank you. We have corrected it to “appeared”.

7.     At line 299 there is a typographical error (should be ‘of’ not ‘auf’).

 Response 7: Thank you, we have corrected this error.

8.     The word ‘legit’ at line 357 should be replaced with a non-slang word, such as ‘legitimate’.

 Response 8: Thank you for this comment. We have changed it to “legitimate”.

Reviewer 2 Report

Line 49-51 - it does not seem as though an online program really offers the opportunity for individualization of care.  I disagree with this premise.  Certainly the availability of internet makes for good rationale for use of this modality, however.

Line 58 poor adherence and high drop outs.  Is there a good reference/studies that show this?  

Line 72 - might be better to use the term evaluate (as opposed to observe)

Lines 85-88 - pretty small N, even for mixed methods/pilot

117-120 - using a shortened version of questionnaires and having only 2 of 10 modules would limit conclusions about usability as well as engagement/adherence.  While this study is specifically focusing on usability, this process seems inadequate to fully test the concepts or the aims of the study

120 - 123.  Would be helpful to have basic info - # questions in the tool, scales/subscales, type of rating scale, reliability information (Cronbach's alpha) - I shouldn't have to go read a whole other paper to be able to judge the tools you are using.

154-157 would be helpful to know reliability stats for the SUS from publisher or other users in some similar study(s)

174 how was the 68 determined to be above average?   Seems like it would be good to reference this from literature or other instrumentation site.

178-180 - the first focus group is not gender balanced and is the opposite distribution of gender from the second group.  This poses a major limitation.

Limitations - addressed many of the above-noted issues, to your credit.

ending - seems abrupt - no summary/conclusion.  Seems like it ought to link back to lines 75 and 76 which talked about usability - 1 or 2 sentence summary, followed by next steps for implementation of the overall program in schools.  I really don't think you addressed the need for assessment of engagement prior to program start.  Could you say something more about this that ties things together better?

Author Response

Response to Reviewer Comments

We would like to thank the reviewers for their valuable comments and suggestions. After completion of the suggested edits, we believe that the manuscript has considerably improved. Please find below a description of how each comment was addressed in the manuscript. (Changes compared to the original submission are highlighted in the manuscript.)

Reviewer 2:

Line 49-51 - it does not seem as though an online program really offers the opportunity for individualization of care.  I disagree with this premise.  Certainly the availability of internet makes for good rationale for use of this modality, however.

Response 1: Thank you for this comment. We agree with your objection and have changed this paragraph. However, online programs do allow some kind of tailoring of program content based on basic characteristics of participants (like gender) or risk status which is especially important when online programs are offered in group settings like school classes.

Line 58 poor adherence and high drop outs.  Is there a good reference/studies that show this?  

Response 2: Thank you for this question. Yes, we have added two references here – Eysenbach (2005) which is a core paper on this topic and Kelders et al. (2012) who have conducted a systematic review including more than 100 papers focusing on adherence of online interventions.

Line 72 - might be better to use the term evaluate (as opposed to observe)

Response 3: Thank you, we have changed it to “evaluate”.

Lines 85-88 - pretty small N, even for mixed methods/pilot

Response 4: Thank you for this comment. We have now added a rationale for the obtained sample size in the methods section. Nielsen et al. found that usability testing with five participants will obtain more than 85% of usability problems and that more participants will mainly produce repetitive information.

117-120 - using a shortened version of questionnaires and having only 2 of 10 modules would limit conclusions about usability as well as engagement/adherence.  While this study is specifically focusing on usability, this process seems inadequate to fully test the concepts or the aims of the study

Response 5: Thank you for this comment. We agree that testing only parts of the program limits the generalizability to those parts that were not tested as stated in the limitation section. The decision to test only parts of the program was also met due to practical reasons and time constraints. A usability test session as described in the manuscript took about 90 minutes in total (including instruction, think aloud task, semi-structured interview and SUS questionnaire). Testing all 10 modules of the program would have required a minimum of 4-5 hours which was not feasible. However, previous research indicates that it is still worth to test the first parts of the program as the start of an online program is known to be crucial for future adherence. A couple of studies (e.g. Eysenbach, 2005; Wangberg et al., 2010; Wanner et al., 2008) demonstrated that most participants drop out from online interventions during the very early stage of a program. Thus, evaluating the program start seems highly relevant. This was also demonstrated in a previous study by Nitsch et al. (2016), whose study design was applied to the current study. We explained this point in more detail in the limitation section.  

120 - 123.  Would be helpful to have basic info - # questions in the tool, scales/subscales, type of rating scale, reliability information (Cronbach's alpha) - I shouldn't have to go read a whole other paper to be able to judge the tools you are using.

Response 6: Thank you for this comment. We have added information concerning the used questionnaires by adding the scales / subscales used, number of items and type of ratings and information regarding reliability.

154-157 would be helpful to know reliability stats for the SUS from publisher or other users in some similar study(s)

Response 7: We have added a reliability (Cronbach Alpha = .91) reported by Bangor et al. (2008) who have considered studies using the SUS questionnaire.

174 how was the 68 determined to be above average?   Seems like it would be good to reference this from literature or other instrumentation site.

Response 8: Thank you for this question. We have now cited a more updated publication reporting how the SUS score should be interpreted (Bangor et al. 2008) and added more information regarding this point. In this paper, a SUS score of > 70 is regarded as acceptable usability. This score refers to the average SUS scores found in other usability studies using the SUS. A SUS score of  > 85 is regarded as excellent usability.

178-180 - the first focus group is not gender balanced and is the opposite distribution of gender from the second group.  This poses a major limitation.

Response 9: Thank you for this comment. We agree that this is a major limitation of the study and therefore we have discussed this point prominently in the limitation section.

Limitations - addressed many of the above-noted issues, to your credit.

ending - seems abrupt - no summary/conclusion.  Seems like it ought to link back to lines 75 and 76 which talked about usability - 1 or 2 sentence summary, followed by next steps for implementation of the overall program in schools.  I really don't think you addressed the need for assessment of engagement prior to program start.  Could you say something more about this that ties things together better?

Response 10: Thank you. We have added a conclusion based on your suggestions.

Round  2

Reviewer 1 Report

This is an excellent manuscript, and all of my concerns were addressed.

Author Response

Thank you very much for your valuable comments.